# *In vivo* dynamics and adaptation of HTLV-1-infected clones under different clinical conditions

**Mikiko Izaki**[1], **Jun-ichirou Yasunaga**[1,2], **Kisato Nosaka**[1], **Kenji Sugata**[2], **Hayato Utsunomiya**[3], **Youko Suehiro**[3], **Takafumi Shichijo**[1], **Asami Yamada**[1], **Yasuhiko Sugawara**[4], **Taizo Hibi**[4], **Yukihiro Inomata**[4], **Hirofumi Akari**[5], **Anat Melamed**[6], **Charles Bangham**[6], **Masao Matsuoka**[1,2]*

1 Department of Hematology, Rheumatology, and Infectious Diseases, Graduate School of Medical Sciences, Faculty of Life Sciences, Kumamoto University, Kumamoto, Japan, 2 Laboratory of Virus Control, Institute for Frontier Life and Medical Sciences, Kyoto University, Kyoto, Japan, 3 Department of Hematology, National Kyushu Cancer Center, Fukuoka, Japan, 4 Department of Transplantation and Pediatric Surgery, Graduate School of Medical Sciences, Faculty of Life Sciences, Kumamoto University, Kumamoto, Japan, 5 Center for Human Evolution Modeling Research, Primate Research Institute, Kyoto University, Inuyama, Aichi, Japan, 6 Section of Virology, Department of Medicine, Imperial College London, London, United Kingdom

* mamatsu@kumamoto-u.ac.jp

**Data Availability Statement:** All data are in the manuscript and its supporting information files.

**Funding:** This research is supported by a grant from the Project for Cancer Research And

## Abstract

Human T-cell leukemia virus type 1 (HTLV-1) spreads through cell contact. Therefore, this virus persists and propagates within the host by two routes: clonal proliferation of infected cells and *de novo* infection. The proliferation is influenced by the host immune responses and expression of viral genes. However, the detailed mechanisms that control clonal expansion of infected cells remain to be elucidated. In this study, we show that newly infected clones were strongly suppressed, and then stable clones were selected, in a patient who was infected by live liver transplantation from a seropositive donor. Conversely, most HTLV-1+ clones persisted in patients who received hematopoietic stem cell transplantation from seropositive donors. To clarify the role of cell-mediated immunity in this clonal selection, we suppressed CD8+ or CD16+ cells in simian T-cell leukemia virus type 1 (STLV-1)-infected Japanese macaques. Decreasing CD8+ T cells had marginal effects on proviral load (PVL). However, the clonality of infected cells changed after depletion of CD8+ T cells. Consistent with this, PVL at 24 hours *in vitro* culture increased, suggesting that infected cells with higher proliferative ability increased. Analyses of provirus in a patient who received Tax-peptide pulsed dendritic cells indicate that enhanced anti-Tax immunity did not result in a decreased PVL although it inhibited recurrence of ATL. We postulate that *in vivo* selection, due to the immune response, cytopathic effects of HTLV-1 and intrinsic attributes of infected cells, results in the emergence of clones of HTLV-1-infected T cells that proliferate with minimized HTLV-1 antigen expression.

Therapeutic Evolution (P-CREATE) (20cm0106306h0005 to M. M.), the Research Program on Emerging and Re-emerging Infectious Diseases (20fk0108088h0002 to M. M.) from the Japan Agency for Medical Research and Development (AMED), JSPS KAKENHI (19H03689 to M.M. and 20H03514 to J-I. Y.). This study was also supported in part by the JSPS Core-to-Core Program A, Advanced Research Networks. The funders had no role in study design, data collection and analysis, decision to publish, or preparation of the manuscript.

**Competing interests:** The authors have declared that no competing interests exist.

## Author summary

HTLV-1 spreads *in vivo* through two routes: *de novo* infection and clonal proliferation of infected cells. Reverse transcriptase inhibitors and integrase inhibitors do not influence the PVL in HTLV-1-infected individuals, indicating that clonal proliferation is dominant to maintain and increase PVL *in vivo* in the chronic phase. It is assumed that the host immune responses are critical factors for clonal proliferation. We found that HTLV-1-infected clones dramatically changed during *de novo* infection whereas the clones in the chronic phase survived long-term after transplantation, indicating that HTLV-1-infected clones are selected for survival *in vivo*. Surprisingly, depletion of CD8$^+$ cells had a small impact on PVL in a STLV-1 infected Japanese macaque, but modified the clonality of infected cells. The cells after depletion of CD8$^+$ cells showed a higher proliferative activity during short-term *in vitro* culture. This study reveals that intrinsic attributes of selected clones contribute to clonal proliferation of infected cells.

## Introduction

Human T-cell leukemia virus type 1 (HTLV-1) is the causative agent of adult T-cell leukemia-lymphoma (ATL) and inflammatory diseases such as HTLV-1 associated myelopathy (HAM)/ tropical spastic paraparesis (TSP) [1–3]. The uniqueness of human T-cell leukemia virus type 1 (HTLV-1) is that it spreads mainly through cell-to-cell contact [4]. Therefore, HTLV-1 increases the number of infected cells by promoting proliferation of infected cells, resistance to apoptosis and escape from host immune surveillance. Viral genes are responsible for clonal proliferation, survival of HTLV-1-infected cells *in vivo* and *de novo* infection.

Among viral genes encoded by HTLV-1, the *HTLV-1 bZIP factor* (*HBZ*) gene plays critical roles in the clonal expansion of infected cells [5]. The HTLV-1 transcriptional transactivator protein Tax is essential for *de novo* infection and anti-apoptosis of expressing cells [6,7]. Tax protein is highly immunogenic and well recognized by cytotoxic T lymphocytes (CTLs) [8,9]. Therefore, infected cells transiently express Tax to minimize expression of the immunogenic Tax protein [10]. On the other hand, both the immunogenicity and the level of expression of HBZ protein are very low [11,12]. HTLV-1-infected cells and ATL cells can express HBZ *in vivo*, which enables the expressing cell to proliferate [13]. Furthermore, *HBZ* RNA is also implicated in proliferation and anti-apoptosis of infected cells [13,14]. The special advantage conferred on the virus by the actions of *HBZ* RNA in both non-malignant HTLV-1-infected cells and ATL cells is that CTLs cannot recognize viral RNA.

Anti-Tax CTLs are typically abundant in HTLV-1-infected individuals [9,15], and it has been suggested that they contribute to the control of infected cells. When T cells are stimulated by cross-linking with anti-CD3 antibody, Tax-expressing T cells increased *in vitro* [16]. This finding indicates that Tax promotes proliferation in concert with T-cell receptor mediated stimulation. However, sporadic Tax expression rather suppresses S phase of expressing cells, suggesting that Tax expression is not associated with promoted cell cycling [10,17]. Thus, it remains unknown whether anti-Tax CTLs control proliferation of infected cells *in vivo*. Although the immunogenicity of HBZ is low [11], there is evidence that CTL responses to HBZ control PVL [18–20]. These observations suggest that CTL responses to HBZ control the proliferation of infected cells.

Thus, clonal proliferation and *de novo* infection are influenced by viral gene expression and host immune surveillance. However, it remains unknown how host immune responses to HTLV-1 control proliferation of infected cells *in vivo*. In this study, we analyzed dynamics of infected cells in various clinical situations, and found that the infected clones present immediately after primary infection were replaced by new clones that persisted for a long time.

Furthermore, we analyzed the effect of CD8$^+$ T-cell depletion and Tax vaccination on the clonality of HTLV-1-infected cells. This study reveals *in vivo* effects of host immune responses on HTLV-1 infected cells.

## Results

### *De novo* infection of HTLV-1 in a patient who received live liver transplantation from a seropositive donor

A seronegative patient with alcoholic hepatitis and liver cancers received live liver transplantation from a HTLV-1 seropositive donor. After two weeks from the transplantation, this patient seroconverted to HTLV-1, and HTLV-1 provirus was detected in the genomic DNA of PBMCs. The proviral load (PVL) was 0.2% at two weeks after transplantation, and increased to 9.7% at one month (Fig 1A). We observed IFN-γ production of T cells when PBMCs were stimulated with Tax peptides using ELISPOT assay, which is similar to those in HAM/TSP patients and a HTLV-1 carrier (Table 1). To analyze the dynamics of HTLV-1-infected clones, integration sites of HTLV-1 provirus were analyzed in sequential samples of PBMCs (1, 3, 6, 12 months) using next-generation sequencing (S1 Table). The 20 most abundant clones present at one month decreased in abundance at three months after transplantation. Conversely, the most abundant 20 clones at 12 months were not detected at one month, but emerged at three months (Fig 1B and 1C). Consistent with this finding, the great majority of clones detected at one month after infection strongly decreased or disappeared (Fig 1D) by three months. Thus, newly-infected HTLV-1 clones dynamically changed, suggesting that appropriate HTLV-1-infected clones for survival *in vivo* are selected. It is noteworthy that many donor-derived HTLV-1 cells persisted in the recipient (Fig 1D).

### HTLV-1-infected clones in ATL patients who received hematopoietic stem cell transplantation from seropositive donors

Next, we analyzed HTLV-1-infected cells in three ATL patients, who received hematopoietic stem cell transplantation (HSCT) from seropositive siblings. Clinical data of these patients were summarized in Table 2. We assume that HTLV-1-infected cells in seropositive donors are already selected to survive *in vivo* in these carriers. We defined the unique integration sites (UIS) detected in the donors as donor-derived clones whereas UIS that is not present in the donor is assumed as newly detected clones. In cases who received HSCT from HTLV-1-seropositive donors, analyses of integration sites showed that a majority of HTLV-1-infected clones detected in the recipients were also found in the donors (abundance of donor-derived clones/ that of total clones: 55.9–81.5%) (Fig 2A and S2 Table). Since these patients were under immunosuppressive condition, reactivation and *de novo* infection of HTLV-1 might occur. Among the clones identified in both donors and recipients, HTLV-1-infected clones with higher abundance in the donor were predominant (Fig 2B). Among the 20 most abundant clones in the recipients (red lines in Fig 2B), 17 (case A), 20 (case B), and 15 clones (case C) respectively were found in both donor and recipient. These findings indicate that HTLV-1-infected clones in the donors have been selected for survival *in vivo*, which contrasts with the rapid decline in abundance of newly-infected clones at one month as shown in Fig 1B.

### Effects of suppressed CD8$^+$ T cells or NK cells on proviral load and clonality

These data suggest that host immune surveillance influences the number and clonality of HTLV-1-infected cells. CTLs play an important role to suppress HTLV-1-infected cells *in vivo*

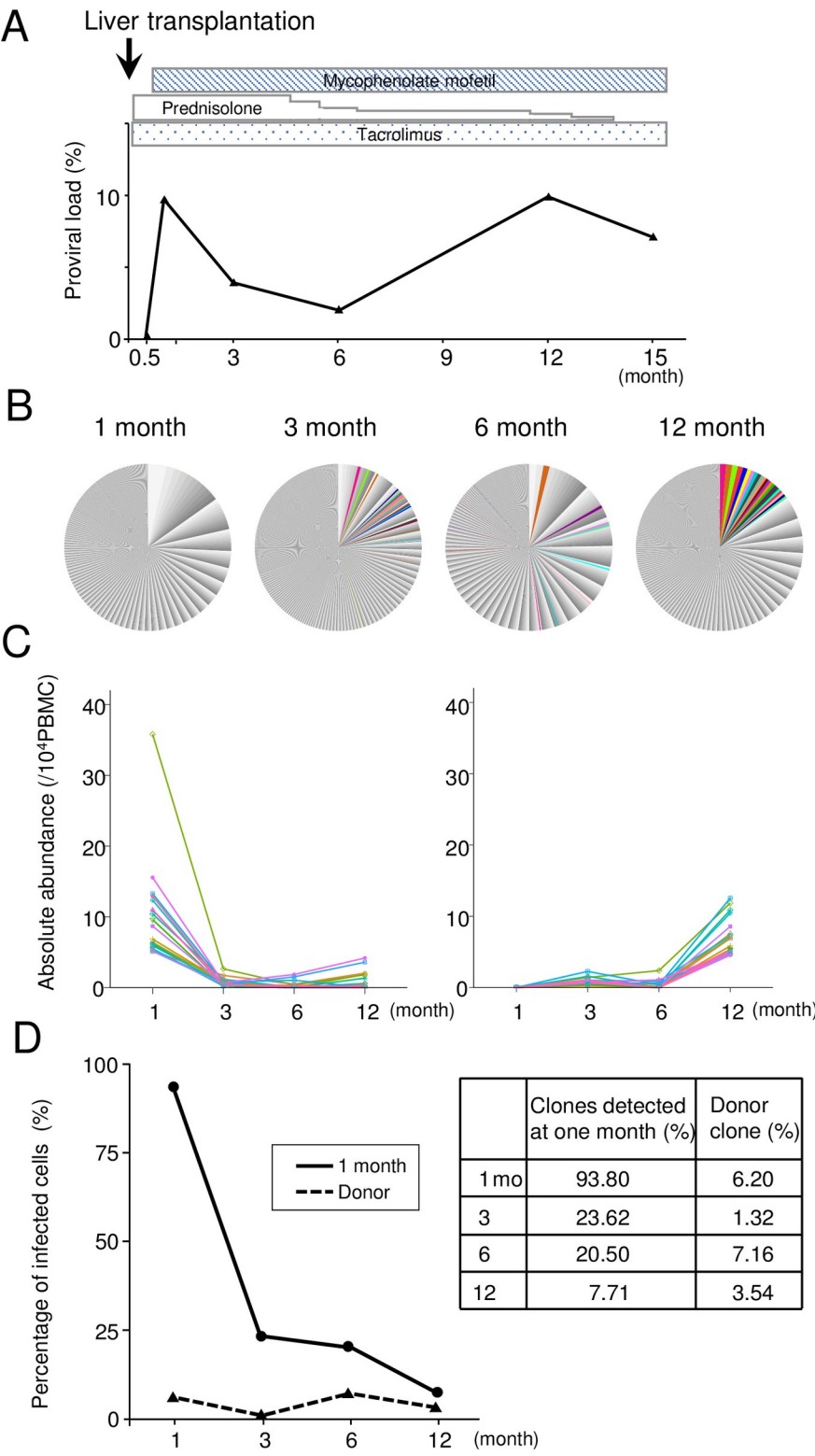

**Fig 1. Proviral load and clonality in the HTLV-1 seronegative patient who received live liver transplantation from a seropositive donor.** (A) Sequential changes of HTLV-1 proviral load (%) after liver transplantation and usage of immunosuppressive agents. (B) HTLV-1 integration sites were determined using high throughput sequencing. The relative frequency of HTLV-1-infected clones is presented. Each area in the pie charts represents the relative frequency of an individual clone (identified by its unique integration site). The top 20 high abundant clones at 12 months are

shown in color. (C) The absolute abundance of the 20 most abundant HTLV-1-infected clones at one (left panel) and 12 months (right panel) is shown. (D) Percentages of infected cells derived from clones of recipient at one month (solid line) and those of donor (dashed line) are shown. The percentages of clones detected at one month in the recipient and the donor are presented.

[21]. Tax is immunodominant in the CTL response to HTLV-1 [9], although the CTL response to HBZ is the more important determinant of the proviral load [18,20]. We hypothesized that depletion of CD8$^+$ T cells would increase the number of infected CD4 T cells *in vivo*. To analyze the effect of CTLs *in vivo*, we used Japanese macaques (JMs) naturally infected with simian T-cell leukemia virus type 1 (STLV-1) as the model for HTLV-1 infection. We have previously reported that STLV-1-infected JMs are similar to HTLV-1-infected individuals in regard to the host immune responses to viral proteins and clonality of virus-infected T cells [22]. First, we administered the monoclonal antibody to CD8 (M-T807R1) to an STLV-1-infected JM. Since this antibody has rhesus constant regions and rhesus variable framework sequences in addition to mouse variable regions, multiple administrations are possible. The proportion of CD8$^+$ T cells was confirmed by flow cytometric analysis. After the first administration of anti-CD8 mAb, CD8$^+$ T cells were strongly suppressed until the fifth week, but recovered at the seventh week (Fig 3A) whereas the PVL in PBMCs remained stable until 12 weeks (Fig 3B). Thereafter, anti-CD8 mAb was administered at 12, 17, and 22 weeks, which led to strong suppression of CD8$^+$ T cells *in vivo* for further 12 weeks (Fig 3A). Accordingly, the PVL gradually increased by 1.39 fold (Fig 3B). Simultaneously, the circulating CD4$^+$ T cell count gradually increased, possibly due to homeostatic proliferation (Fig 3C). As a consequence, the relative frequency of HTLV-1 infection within CD4$^+$ T cells was relatively stable (Fig 3D). Thus, depletion of CD8$^+$ T cells resulted in an increase in the absolute number of infected cells, although the relative ratio of infected cells per CD4$^+$ T cells remains stable.

Next, we analyzed clonality of STLV-1-infected cells *in vivo* using next generation sequencing (S3 Table). As shown in Fig 3E, the 20 most abundant clones (blue lines) present at 24 weeks began to emerge as the dominant clones after 12 weeks, whereas the top 20 clones (red lines) present at 0 weeks were largely suppressed. These data show that the clonality of STLV-1-infected cells dramatically changed after depletion of CD8$^+$ T cells, suggesting that the host immune response influences the clonality of infected cells *in vivo*.

Another type of cell that controls virus-infected cells *in vivo* is the natural killer (NK) cell. To test the effect of NK cells, we administered anti-CD16 monoclonal antibody to a STLV-1-infected JM. Although the number of NK cells was suppressed (S1A Fig), the PVL was relatively stable after treatment (S1B Fig). Since anti-CD16 antibody is xenogeneic, we could

**Table 1. HTLV-1 Tax specific IFN-γ responses in the live-liver transplanted patient, HAM/TSP patients and a HTLV-1 carrier.**

|  | Pt |  | HAM-1 | HAM-2 | HAM-3 | HAM-4 | HAM-5 | Carrier |
|---|---|---|---|---|---|---|---|---|
|  | 3 mo | 12 mo |  |  |  |  |  |  |
| Tax-1 | 0 | 10 | 0 | 0 | 0 | 1 | 291 | 156 |
| Tax-2 | 16 | 3 | 0 | 0 | 0 | 105 | 0 | 0 |
| Tax-3 | 2 | 242 | 3 | 1 | 1 | 0 | 0 | 5 |
| Tax-4 | 0 | 2 | 0 | 0 | 0 | 19 | 0 | 19 |
| Tax-5 | 1 | 0 | 0 | 15 | 0 | 0 | 0 | 0 |
| Tax-6 | 0 | 0 | 0 | 3 | 1 | 4 | 1 | 1 |
| Tax-7 | 0 | 116 | 60 | 68 | 22 | 100 | 2 | 47 |

mo: month

**Table 2. Clinical data of allogeneic hematopoietic stem cell transplantation from seropositive donors.**

|  | Case A | Case B | Case C |
|---|---|---|---|
| **Age** | 40s | 40s | 50s |
| **Sex** | Male | Female | Female |
| **Clinical subtype** | Acute | Acute | Acute |
| **Disease before HSCT** | CR | CR | CR |
| **Donor** | Seropositive sibling | Seropositive sibling | Seropositive sibling |
| **Source of stem cells** | PBSC | Bone marrow | PBSC |
| **HLA compatibility** | Matched | Matched | Matched |
| **Conditioning** | CY/TBI | Flu/Mel | CY/TBI |
| **GVHD prophylaxis** | Cyclosporine-based | Cyclosporine-based | Cyclosporine-based |
| **PVL in donor PBMC (%)** | 1.1 | 7.5 | 0.9 |
| **Cell number of CD34$^+$ cells** | 3.21 x 10$^8$ | Not assessed | 1.79 x 10$^8$ |

CR: complete response, PBSC: peripheral blood stem cell, CY: cyclophosphamide, TBI: total body irradiation, Flu: fludarabine, Mel: melphalan, GVHD: graft-versus-host disease

administer this antibody only once. Repetitive administration is possible for anti-CD8 antibody since it has rhesus constant regions and rhesus variable framework sequences. Therefore, it is difficult to conclude that suppression of NK cells does not influence PVLs in a STLV-1 infected JM. These results suggest that CTLs have mild effects on the total number of HTLV-1-infected T cells in the host, and influence HTLV-1-infected T-cell clones with specific attributes.

Tax expression is suppressed *in vivo* in most infected cells at a given instant; it appears to be expressed in intense intermittent bursts [10,17]. Short-term culture *in vitro*, after removal of CD8$^+$ T cells, induces Tax expression [23]. We analyzed the STLV-1 Tax ortholog sTax expression in CD4$^+$ T cells derived from an STLV-1-infected JM. As well as HTLV-1 carriers, sTax expression was enhanced after depletion of CD8$^+$ T cells (Fig 4A). Contrary to our expectation, the relative expression level of sTax per provirus after 24 h *in vitro* was gradually suppressed after *in vivo* administration of anti-CD8 monoclonal antibody (Fig 4A). The transcription of *tax* is inversely correlated with that of *HBZ* [24]. Therefore, we analyzed transcripts of *STLV-1 bZIP factor* (*SBZ*), which is analogous to HBZ. The expression of *SBZ* did not change after *in vitro* culture except at 0 week (Fig 4B).

It is well known that *in vitro* culture of PBMCs from HTLV-1-infected individuals causes spontaneous proliferation [25]. Both immune responses to viral antigens and proliferation of infected cells are implicated in this spontaneous proliferation. We assayed the PVL before and after *in vitro* culture of PBMCs. After the first administration of anti-CD8 antibody, the proviral load did not increase after *in vitro* culture (Fig 4C). A similar observation was made in other two STLV-1-infected JMs (Fig 4D). However, STLV-1 PVL increased 2.8 times at 25 weeks, suggesting that infected T cells prone to proliferate *in vitro* increased after long-term depletion of CD8$^+$ T cells (Fig 4C), which might reflect the changes observed in the infected clones (Fig 3E). Another possibility is that uninfected T cells cause cell death during the short time culture.

## Dynamics of HTLV-1-infected clones in patients who received Tax peptide pulsed dendritic cell vaccine

Tax is the immunodominant target of CTLs *in vivo* [9]. It has been reported that a Tax peptide-pulsed dendritic cell vaccine enables to maintain remission state of ATL [26,27]. It is

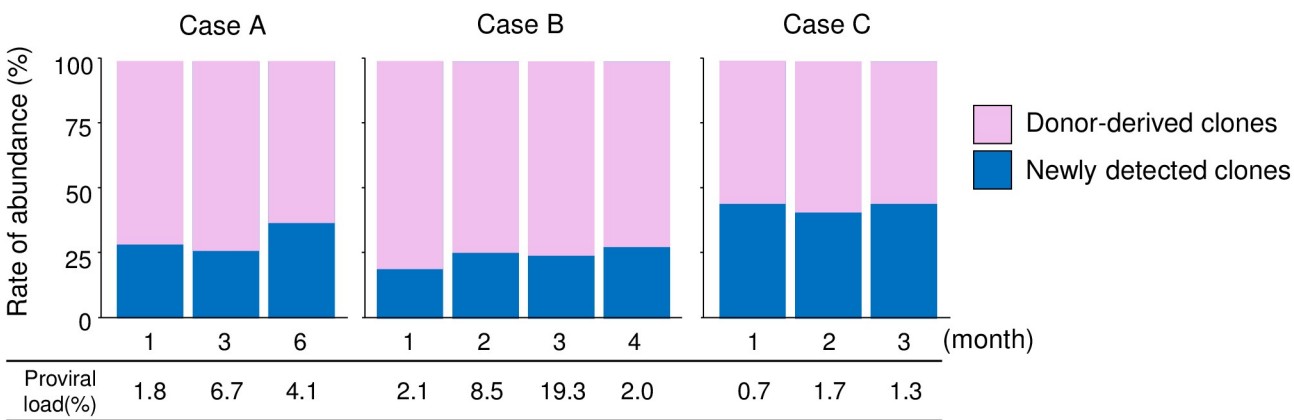

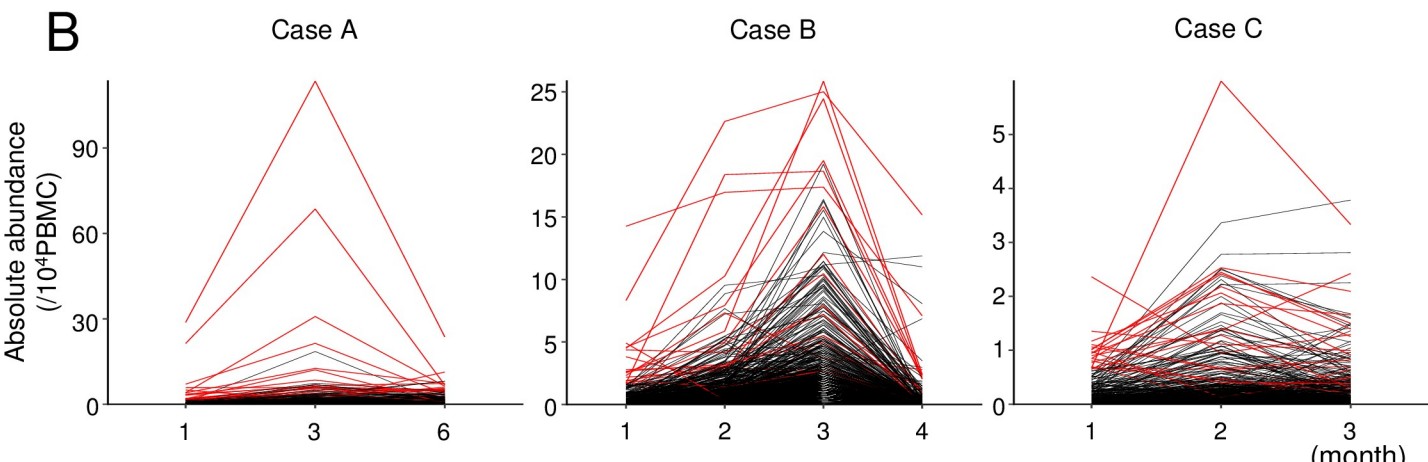

**Fig 2. HTLV-1-infected clones in ATL patients who received hematopoietic stem cell transplantation from seropositive donors.** (A) The frequencies of HTLV-1-infected clones that were detected in the donor sample (pink) and clones newly detected in the recipients (blue) are shown for each observation point. (B) Dynamic changes in abundance of infected clones in recipients. The top 20 abundant clones at first observation point are shown in red.

possible that Tax-expressing non-malignant infected cells are also suppressed by anti-Tax immunity in addition to ATL cells. If so, HTLV-1-infected clones without Tax expression might show a growth or survival advantage *in vivo*. A patient was treated with Tax peptide-pulsed DCs as reported previously [26]. As shown in Fig 5A, PVL remained over 10% after vaccination. Although the PVL changed after the vaccination, it remained high after four months (Fig 5A). Consistent with this finding, most HTLV-1-infected cell clones did not change significantly in abundance (Fig 5B and S4 Table). Since it is possible that Tax non-expressing clones are selected after the vaccination, we analyzed the frequencies of nonsense mutations of the *tax* gene and deletion of 5' long terminal repeat (LTR) [28]. Nonsense mutations (W56* and W248*) in the *tax* gene are frequently detected in ATL cases [29]. As shown in Fig 5C, the frequency of this nonsense mutation (W56*) rather decreased. We have previously reported that deletion of the 5'LTR and internal regions of the provirus is frequently

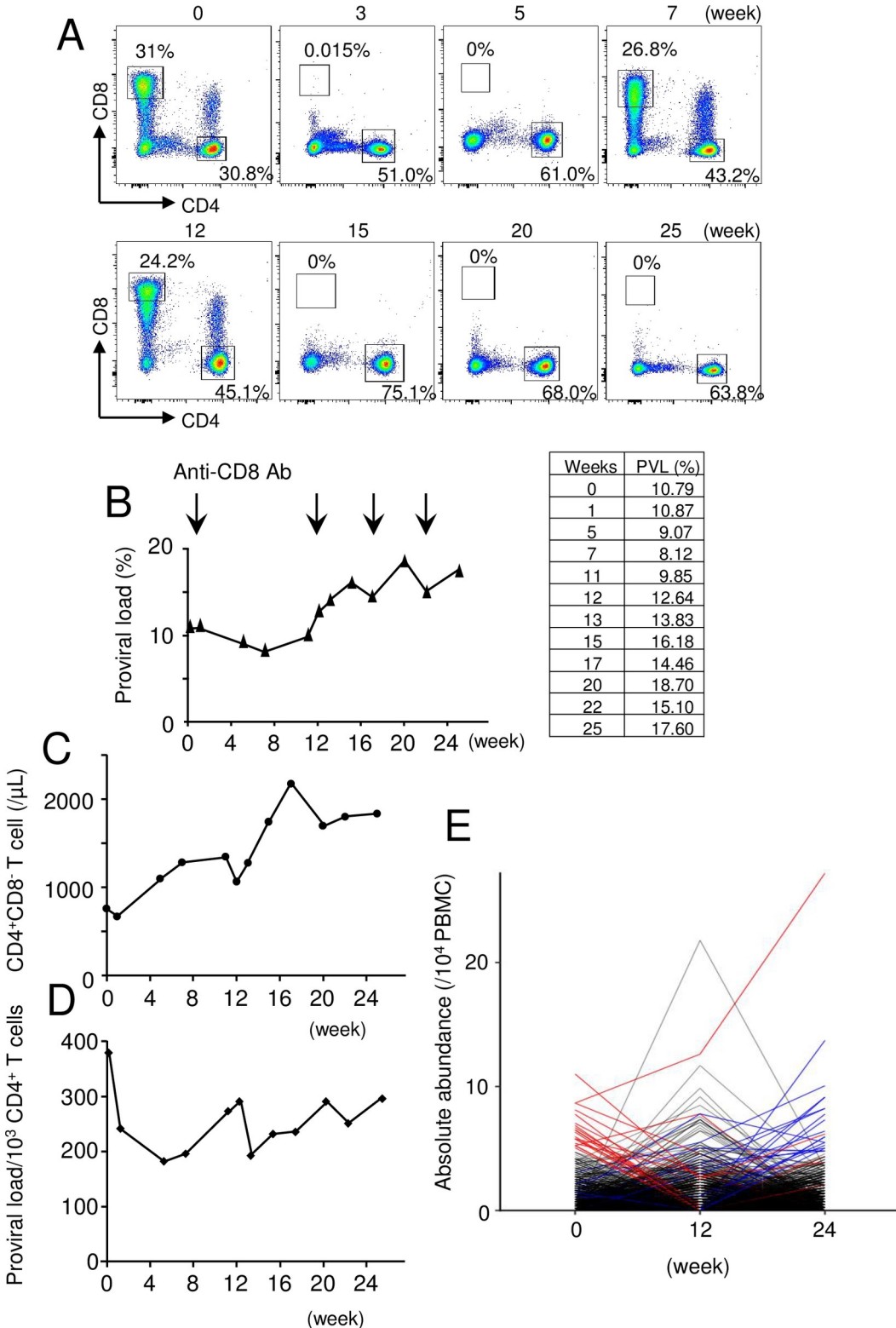

**Fig 3. Effects of anti-CD8 antibody to proviral load and clonality in a STLV-1 infected Japanese macaque.** Anti-CD8 antibody was administered into an STLV-1 infected Japanese macaque. (A) Percentages of CD4+ and CD8+ T cells were analyzed for PBMCs from the infected Japanese macaque before and after administration of anti-CD8 antibody. (B) Proviral load of PBMCs is shown. The time of administration of antibody is indicated by the arrows. (C) The number of CD4+ T cells

per μl is shown in the upper panel. (D) PVL as a proportion of CD4+ T cells instead of PBMCs. (E) Changes in abundance of STLV-1-infected clones is shown. The top 20 abundant clones before administration of antibody are shown in red and those at 24 weeks are shown in blue.

observed in ATL patients, which is designated as type 2 defective provirus [30,31]. 5'LTR contains the promoter/enhancer of transcription of the plus-strand of HTLV-1, so *tax* is not transcribed in type 2 defective viruses. Type 2 defective proviruses have also been found in asymptomatic HTLV-1 carriers [31]. We quantified 5'LTR and the *tax* gene using digital PCR and estimated the frequency of type 2 defective provirus. After vaccination, the frequency of type 2 defective provirus was slightly increased. Collectively, these data show that Tax vaccination had no material effect on the proportion of Tax non-expressing clones (nonsense mutation + type 2) *in vivo*.

## Discussion

HTLV-1 is a unique retrovirus since it transmits only through cell-to-cell contact [4]. After infection, HTLV-1 is present as the provirus in infected cells, and increases the number of infected cells through the functions of two viral genes, *HBZ* and *tax* [4]. Since HBZ and Tax are exogenous antigens, the host immune system functions to eliminate HTLV-1-infected cells. This presents the critical dilemma for persistence of infected cells: excess production of viral antigens induces a strong immune response, whereas their expression is essential to support survival and proliferation of infected cells. As shown in this study, most newly-infected, initially abundant clones disappeared or decreased in abundance, possibly due to the host immune response, after *de novo* infection of HTLV-1. Seroconversion and dynamic changes of clonality were also reported in transplanted patients from the seropositive donor [32]. Similarly, massive deletion of newly-infected clones was also observed in *de novo* infection with bovine leukemia virus (BLV) [33]. The BLV provirus at primary infection was preferentially integrated into the transcribed regions of the genome, but this preference was diminished in the clones that persisted long-term, suggesting that excess production of viral antigens results in rapid exclusion of infected cells. This observation contrasts with the present finding that the infected clones from a seropositive donor persisted in the recipient (Fig 1D). These findings indicate that the infected clones surviving in the donor had already adapted to the *in vivo* environment. However, it remains unknown how the host immune system is implicated in this adaptation. Another possibility is cytopathic effects of HTLV-1 infection. Transient Tax burst induces senescence of expressing cells [34]. In addition, Tax expression induces Fas ligand expression, leading to activation-induced cell death [35]. Tax also induces genetic instability through inhibition of homologous recombination repair [36]. Thus, HTLV-1 infected cells that can reduce cytopathic effects of viral proteins are likely selected *in vivo*. HBZ and Rex are implicated in suppression of Tax expression and cytopathic effects [37]. Taken together, the host immune surveillance and cytopathic effects of HTLV-1 infection are implicated in selection of adapted HTLV-1 infected cells.

Since CTLs suppress HTLV-1-infected cells *in vivo*, it is assumed that their depletion allows infected cells to increase. Depletion of CD8+ T cells in an STLV-1 infected JM had little impact on PVL in the first 12 weeks, although clonal changes in STLV-1 infected cells were already evident by this time (Fig 3E). Since CTLs target peptides derived from viral proteins, the total abundance of HTLV-1 CTL target antigens, not the PVL, that is the object of CTL-mediated selection. Therefore, it is noteworthy that clonality has changed after deletion of CD8+ T cells. This suggests that infected T-cell clones that are suppressed by CTLs increase in number and abundance *in vivo*. Interestingly, short-term culture of CD4+ T cells results in increased PVL

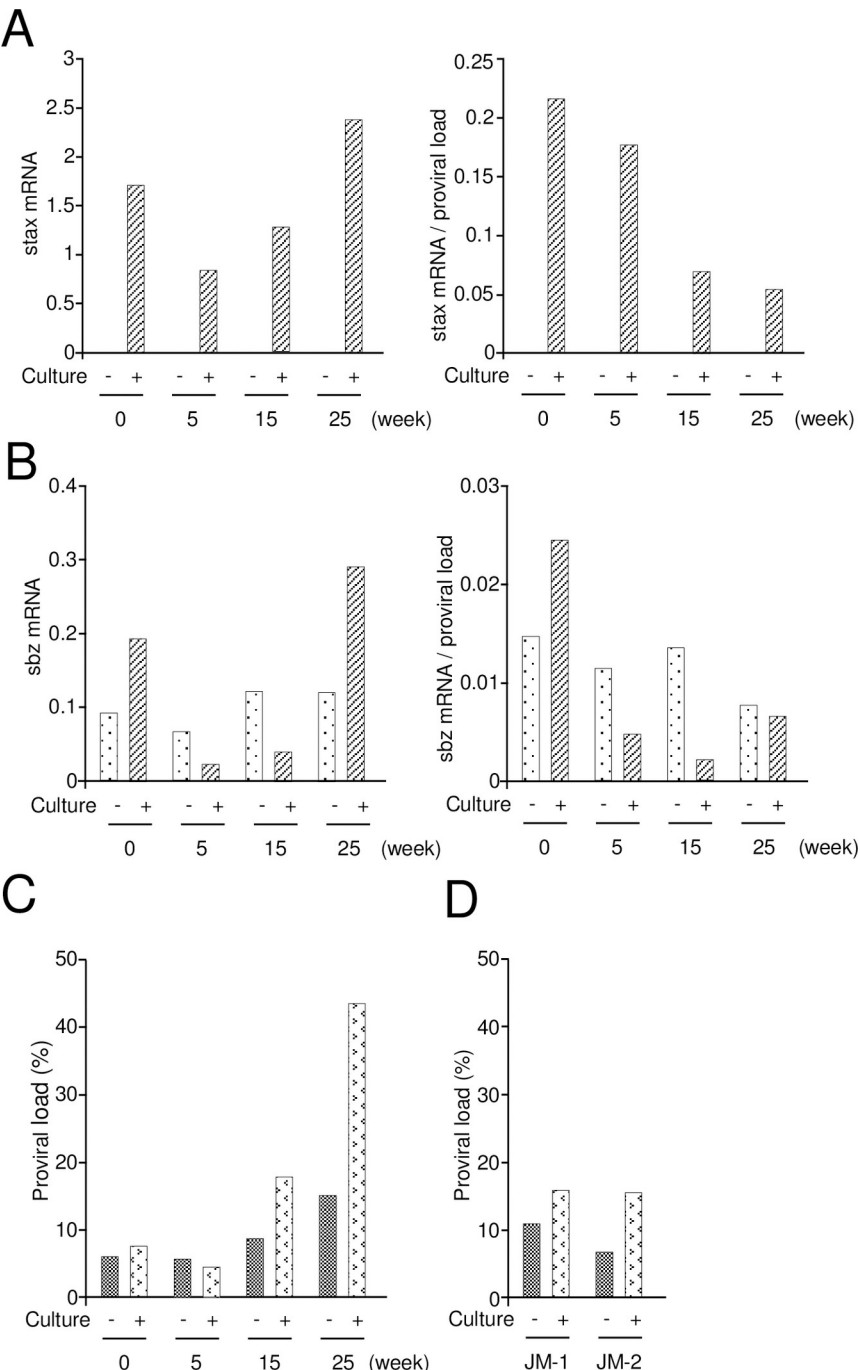

**Fig 4. *stax* and *sbz* expression, and proviral load in the CD8⁺-depleted Japanese Macaque.** The expression level of *stax* and *sbz* in CD8 depleted Japanese macaque was measured by quantitative PCR, with (striped) or without (dotted) *in vitro* culture (24 hrs). Expression of CD4⁺CD8⁻ T cells is shown in the left. The relative expression level per provirus is shown in the right: (A) *stax* transcripts, (B) *sbz* transcripts. (C) PVL before and after *in vitro* culture was measured in each sample. (D) PVLs were measured in two STLV-1 naturally infected Japanese macaques (JM-1, 2) without anti-CD8 antibody as control.

after depletion of CD8⁺ T cells, indicating that infected T cells that are prone to proliferate are predominant. These findings suggest that intrinsic attributes contribute to the increased number of infected cells. What is the molecular mechanism that underlies this phenomenon? It is

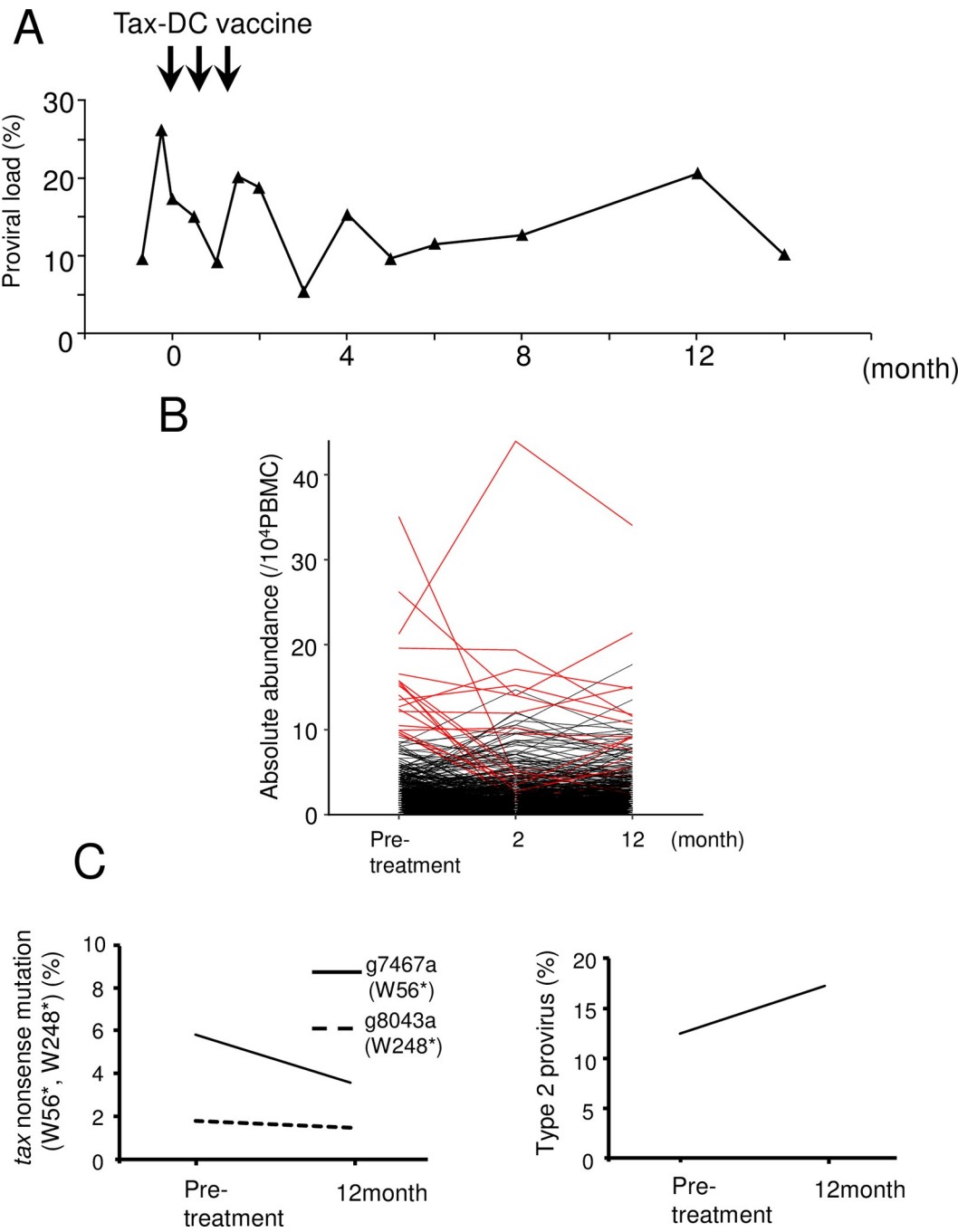

**Fig 5. Dynamics of HTLV-1-infected clones in a patient who received Tax peptide-pulsed dendritic cell vaccine.** (A) Proviral load (%PBMCs) before and after treatment with a Tax peptide-pulsed dendritic cell vaccine. The time of vaccination is indicated by the arrows. (B) Dynamics of the abundance of infected clones in this treated patient. The top 20 abundant clones before treatment are shown in red. (C) The frequency of the infected clones that have *tax* nonsense mutation (W56*, W248*) at each observation point is shown in left panel. The frequency of type 2 provirus (lacking 5'LTR and internal regions of provirus) is shown in the right panel.

possible that integration sites influence expression pattern of viral genes, and suitable infected cells for proliferation with minimized viral gene expression are selected. Recently, transient Tax expression in ATL cell lines and HTLV-1-infected cell lines was reported [10,17]. Tax

aberrantly activates T cells through triggering of the T-cell receptor complex [38]. Further-more, T-cell activation enhances proliferation of Tax-expressing cells in LTR-tax transgenic mice [16]. Thus, transient Tax expression might facilitate infected T cells to proliferate *in vivo* although Tax expression in total does not increase. Furthermore, the proliferative ability of HBZ-expressing T cells is greater than that of normal T cells [13]. Another intrinsic attribute of HTLV-1-infected cells of possible relevance is impact of the CTCF binding site in the provirus [39]. The HTLV-1 provirus forms loop structures with CTCF binding sites in the host genome, which influences transcription of host genes [40]. HTLV-1-infected clones with altered host gene expression that promotes *in vivo* proliferation might be selected.

Another possibility of changed clonality is the cytopathic effect of HTLV-1. As described above, high level of Tax expression induces apoptosis, senescence and suppresses cell cycle [34–36]. Depleted CD8 T cells likely leads to enhanced Tax expression and cytopathic effects. This is supported by the findings that stax mRNA/STLV proviral DNA decreased after deple-tion of CD8$^+$ T cells since Tax overexpressing cells decrease by cytopathic effects. In addition, significantly increased PVLs after depletion of CD8$^+$ T cells is that uninfected cells cause apoptosis probably due to infected cells. Fas ligand might be involved in this mechanism.

The limitation of this study is that only one STLV-1 infected JM was analyzed. Since animal resources were limited, we selected an STLV-1 infected JM with high proviral load. It is possible that depletion of CD8$^+$ T cells causes different changes of PVLs in other JMs. Depletion of CD8$^+$ T cells *in vitro* induces increased *tax* transcripts (Fig 4A), which is consistent to findings in human samples [23]. However, *in vivo* effects of depleted CD8$^+$ T cells is different from those *in vitro*. Therefore, this study reveals new findings regarding *in vivo* effects of CD8$^+$ T cells.

Tax peptide-pulsed dendritic cells can suppress leukemic cells in ATL patients [26]. However, the HTLV-1 PVL increased in a patient who underwent complete remission after the vaccination, which suggests that the PVL does not reflect the disease response of Tax vaccination [26]. In line with this finding, HTLV-1 PVL was not suppressed in the present patient. However, this patient remained in complete remission after the vaccination, indicating that Tax vaccination controlled ATL. These findings suggest that the immune response to Tax does not control the number of infected cells (PVL), but suppresses ATL. It is thought that these ATL cases are Tax-dependent. The conclusion that the CTL response to Tax has little impact on the proviral load is consistent with the previous observations that *tax* is expressed in uncommon, intermittent bursts [10,17], and that the HLA class 1-restricted immune response that correlated with control of the PVL is the response to HBZ [18,19], which is expressed in ~50% of HTLV-1-infected PBMCs at a given instant [41].

Tax is not expressed in approximately half of ATL cases [42]. Three mechanisms inactivate Tax expression: 1) nonsense mutation of the *tax* gene [29,43], 2) deletion of the 5' LTR [30,31], and 3) DNA methylation of the 5'LTR [44,45]. Tax can be expressed in the remaining half of ATL cases, in which the *tax* gene does not contain nonsense mutations or deletions, and the 5'LTR is neither deleted nor methylated. However, since the level of *tax* transcripts is very low by RT-PCR, it remains unknown whether these ATL cells indeed express Tax. Recently, we have reported that a fraction of ATL cells transiently express Tax, which is critical to maintain the whole ATL cell population [10]. As a mechanism, transient Tax expression activates transcription of genes important for anti-apoptosis, which generates vigorously proliferating cells [10]. It is thought that ATL cases that retain Tax-expressing capability depend on transient Tax expression. Therefore, an enhanced CTL response to Tax induced by vaccination likely suppresses transient Tax expression, leading to suppression of ATL cells.

In this study, we have shown both dynamic changes and stability of HTLV-1-infected cell clones in transplanted patients. During the chronic phase of HTLV-1 infection, *in vivo*

selection, possibly by the immune response, results in the emergence of long-lived clones with particular intrinsic attributes. The detailed mechanisms should be elucidated in the future.

## Materials and methods

### Ethics statement

Blood samples from ATL patients and HTLV-1 carriers were collected after the written informed consent was obtained in accordance with the Declaration of Helsinki. These experiments were approved by the Institutional Ethics Committee of Kumamoto University (G374). Blood sample from patient treated with Tax-dendritic cell vaccine were provided from National Hospital Organization Kyushu Cancer Center.

### Statement of animal ethics

Four Japanese monkeys (*Macaca fuscata*) were used for this study. All monkeys were supplied from colonies in the Primate Research Institute. The monkeys were reared in outdoor group cages with wooded toys provided as environmental enrichment. They were fed with apple, potato and commercial monkey diet. They were able to access to water ad libitum. They had own health record from birth with yearly health checkup. Blood samples were obtained from the macaques under ketamine anesthesia with medetomidine, followed by administration of its antagonist atipamezole at the end of the procedure. At euthanasia, ketamine anesthesia to the macaques was followed by injection of pentobarbital sodium at a dose of $\geq$25 mg/kg. Then they were perfused with phosphate buffered saline (PBS) to get rid of the contamination of blood cells in solid organs before necropsy for this study. The animal experiment was approved by the Animal Welfare and Animal Care Committees of Kyoto University (approval number R11-11, R12-01, R13-01, R14-01 and R15-01), and was carried out in accordance with the Guidelines for Care and Use of Nonhuman Primates (Version3) by the Animal Welfare and Animal Care Committee of KUPRI. This guideline was prepared based on the provisions of the Guidelines for Proper Conduct of Animal Experiments (June 1, 2006; Science Council of Japan) as well as Fundamental Guidelines for Proper Conduct of Animal Experiment and Related Activities in Academic Research Institutions [Notice No. 71 of the Ministry of Education, Culture, Sports, Science and Technology dated June 1, 2006], in accordance with the recommendations of the Weatherall report, "The use of non-human primates in research": http://www.acmedsci.ac.uk/more/news/the-use-of-non-human-primates-in-research/.

### Human samples and Japanese macaque samples

Human blood samples were obtained from the following patients. A seronegative patient had alcoholic cirrhosis (non-compensated state) and hepatocellular carcinoma. He received a living liver transplant of a seropositive donor. Immunosuppressive therapy after transplantation includes tacrolimus, prednisolone after transplantation, and mycophenolate mofetil was started one month after transplantation. Prednisolone was tapered and was terminated at 14th month after transplantation. Three patients with acute ATL achieved complete remission by chemotherapy prior to transplantation. Case A received a peripheral blood stem cell transplantation from the elder sister. Case B received a bone marrow transplantation from the younger brother. Case C received a transplantation from the younger brother. All donors were seropositive. The transplantations were successful and full donor chimerism was observed in all recipient's bone marrow. An ATL patient was treated with Tax peptide pulsed dendritic cell vaccine.

PBMC were isolated using density gradient media; Ficoll-Paque (GE Healthcare Bio-Science).

## Treatment of STLV-1 infected Japanese macaques with anti-CD8 antibody or anti- CD16-antibody

A Japanese macaque infected with STLV-1 was treated with anti-CD8 antibody M-T807R1 obtained from the Nonhuman Primate Reagent Resource funded by the National Institutes of Health. M-T807R1 contains mouse variable regions for CD8α and rhesus constant regions and rhesus variable framework sequences. Therefore, multiple administration is possible for this antibody. The antibody was administered subcutaneously (50mg/kg) at 0, 12, 17, and 22 weeks. Before each administration, a 10 ml of blood sample was obtained. Until week 25, blood samples were collected as following weeks; week 1, 3, 5, 7, 11, 13, 15, 17, 20, 22, 25. Another STLV-1 infected Japanese macaque treated with anti-CD16 antibody 3G8 (BD Biosciences or DJ130c, Dako, Glostrup, Denmark). Since this is a mouse antibody, we can administer this antibody only once. Antibody solutions in phosphate-buffered saline (PBS) are administered by slow intravenous bolus injection at 50mg/kg on day 0. Blood samples were obtained before administration and following day; day 3, 8, 10, 14, 17, 21, 35, 49.

## Proviral load

Genomic DNA was extracted with phenol/chloroform method as previously described [46]. PVL was measured by real-time PCR. The copy number of the *tax* region and RAG1 gene in genomic DNA was quantified. HTLV-1 PVL was calculated with relative quantification method by using TL-Om1 of which PVL is 100%. STLV-1 PVL was calculated with absolute quantification method. The sequences of primers of RAG-1 and *tax* were as follows; *tax* primer (human) 5'-GAAGACTGTTTGCCCACCACC-3' (sense) and 5'-TGAGGGTTGAGTGGAA CGGA -3' (anti-sense); *tax* probe (human) 5'-CACCCGTCACGCTAACAGCCTGGCAA-3'; RAG-1 primer (human) 5'-CCCACCTTGGGACTCAGTTCT-3' (sense) and 5'-CACCCGGA ACAGCTTAAATTTC-3' (anti-sense); RAG-1 probe (human) 5'-CCCCAGATGAAATTCA GCACCCACATA-3'; *tax* primer (simian) 5'- CTACCCTATTCCAGCCCACTAG-3' (sense) and 5'- CGTGCCATCGGTAAATGTCC-3' (anti-sense); *tax* probe (simian) 5'- CACCCGCC ACGCTGACAGCCTGGCAA-3'; RAG-1 primer (simian) 5'- CCCACCTTGGGACTCAG TTCT-3' (sense) and 5'- CACCCGGAACAGCTTAAATTTC-3' (anti-sense); RAG-1 probe (simian) 5'- CCCCAGATGAAATTCAGCACCCATATA-3'. All probes were labeled with fluorescent 6-carboxyfluorescein (FAM) (reporter) at the 5' end and fluorescent 6-carvoxy tetramethyl rhodamine (TAMRA) (quencher) at 3' end. The reaction conditions were 50°C for 2 min, 95°C for 10 min and 45 cycles of 15 seconds at 95°C, followed by 60 seconds at 60°C.

## Quantitative analysis of viral gene expression

CD8+ T cells were removed by positive selection with BD IMag (BD Bioscience) from PBMC of STLV-1 infected Japanese macaque. CD8+ T depleted cells were cultured for 24 hours with RPMI 1640 medium supplemented with 10% fetal bovine serum (FBS) and antibiotics. Total RNA was extracted from pre-cultured and post-cultured CD8 T deleted cells and Si-2, which is a STLV-1-infected T cell line derived from JM, using Trizol reagent (Thermo Fisher Scientific). cDNA was synthesized with SuperScript IV (Thermo Fisher Scientific) using random primer. The transcripts of tax and those of SBZ were detected by real time PCR. As internal control, GAPDH mRNA was measured. The primers and probes were as follows; stax primers; 5'- ATCCCGTGGAGGCTCCTC-3' (sense) and 5'- CCAAATACGTAGACTGGGTATCCA T-3' (anti-sense); stax probe; 5'- ACCAACACCATGGCCCACTTCCC-3'; SBZ primers; 5'- A GAGCGCAACTCAACCGG-3' (sense) and 5'- GCAGGGAACAGGTAAACATCG-3' (anti-sense); SBZ probe; 5'- TGGATGGCGGCCTCAGGGCC-3'.; GAPDH primers; 5'- ACCAAC

TGCTTAGCACCCCT-3'(sense) and 5'- GTCTTCTGGGTGGCAGTGAT-3'(anti-sense). The expression of stax and SBZ were measured by TaqMan realtime PCR. GAPDH transcripts were measured by SYBR-green real time PCR. The amplification conditions were 50˚C for 2 min, 95˚C for 10 min and 45 cycles of 15 seconds at 95˚C, followed by 60 seconds at 60˚C. The relative expression levels of tax and SBZ were quantified by ddCt method using Si-2 as a reference.

## Flow cytometric analysis

The antibodies used for immunophenotyping the PBMCs of Japanese macaques were as follows: anti-CD4 (OKT4)(BioLegend), anti-CD8 (DK25)(Dako), anti-CD3 (SP34-2)(BD Bioscience), anti-CD159A (NKG2A, A199)(Beckman Coulter). Anti-CD8 antibody can detect CD8+ T cells in the presence of M-T807 antibody [47]. To avoid blocking of CD16 detection by the administered antibody, NK cells were defined as CD3-CD159A+ lymphocytes [48]. Samples were analyzed by a FACSVerse with FACSuite software (BD Biosciences) and data was analyzed with Flow Jo software (FlowJo, LLC).

## High throughput sequencing of provirus integration sites and tax nonsense mutation

For analyses of clonality, the provirus integration site in human and the Japanese macaque genome were amplified by linker-mediated PCR as previously described with some modification using Miseq (Illumina)[22,46,49]. Genomic DNA was sheared by sonication with a Covaris S220 instrument (Covaris) to obtain DNA fragments of approximately 200–500 bp. After end-repair and linker ligation, nested PCR was performed to amplify the integration sites. The primers of specific for viral and linker sequences were used. Amplicons were ligated with the adaptor specific for Miseq using TruSeq DNA PCR-Free Smaple Prep Kit (Illumina). PCR products after nested PCR were used as input DNA. High throughput sequencing was performed according to the manufacturer's instructions.

To check the presence of *tax* nonsense mutation, HTLV-1 sequence was amplified by PCR. The primers were as follows; 5'- TACGTCTTTGTTTCGTTTTCTGTTCTCGCCG-3' (sense) and 5'- AGAGCCGGCTGAGTCTAGGTAGGCT-3' (anti-sense). The amplification conditions were 35 cycles of 10 seconds at 98˚C, followed by 10 minutes at 68˚C. PCR products were ligated the adaptor by using Nextra XT DNA Sample Preparation Kit (Illumina). Then high throughput sequencing was performed with MiSeq (Illumina).

## Bioinformatics

The obtained reads were trimmed with a quality threshold of 20 on the Phred scale in order to remove low quality reads using Trim Galore (http://www.bioinformatics.babraham.ac.uk/projects/trim_galore/). The host genomic sequences adjacent to the viral 3' LTR ("GCACTCTCAGGAGAGAAATTTAGTACACA" was used as a marker for HTLV-1, "CTCTCTCCAGGAGAGAGGTTTAGTACACA" was those of STLV-1) and the linker sequence (CCTCTCTATGGGCAGTCGGTGATCGCTCTTCCGATCT) was used as a marker) were extracted from the reads using Cutadapt software package. Those trimed reads were arranged as Rread1 including sequences started from the beginning base of integration site and Read2 including sequences started from the end base of shear-site using Cutadapt software package. Trimmed reads were aligned to human genome reference (UCSC hg38) or *Macaca mulatta* genome reference (Mmul_8.0.1) using Burrows-Wheeler Aligner (BWA). The reads were filtered by mapping quality, removing supplementary reads, and excluding un-paired reads and reads of virus sequence with SAMtools software package. Minus strand sequences were converted into

complementary sequencing to count the number of clones and PCR duplicates. The number of integration site was calculated by counting the number of amplicon with different shear site. Sequence similarity for longest sequence in each integration site was evaluated through the program ClustalW (version2) in order to remove twin integration sites arising from mismapping of some duplicates. With regard to the pair of clones with high homology score (>85), the clone which has the smaller number of shear site was removed. When the number of shear site was same, we used total read number including the number of PCR products in addition to the number of shear sites. Furthermore, both clones were excluded when the pair of clones has the same shear-sites number and same number of reads. The absolute abundance per 10000 PBMCs of given unique integration site was calculated by using number of UIS and proviral load. We defined all UIS detected in donors as donor clone. In recipient, we defined the UIS other than donor clones as newly detected clones. For tax nonsense mutation, the obtained data were analyzed by CLC Genomics Workbench (ver. 10.1.1, CLC bio).

## HTLV-1 Tax Peptide

Overlapping peptides of nine amino acids in length (offset: 1amino acids) were designed based on the amino acid sequences of Tax (Genbank accession number: AB513134). 345 Tax peptides were pooled as Tax-1 ($Tax_{1-50}$), Tax-2($Tax_{51-100}$), Tax-3($Tax_{101-150}$), Tax-4 ($Tax_{151-200}$), Tax-5 ($Tax_{201-250}$), Tax-6 ($Tax_{251-300}$), Tax-7 ($tax_{301-345}$).

## Enzyme-linked immunosorbent spot (ELISPOT) assay

PBMCs of the patient who received a live liver transplantation were subjected to ELISPOT assay with human IFN-γ ELISPOT kit (MABTECH). Cells were seeded into ELISPOT plates and stimulated with 1 μM pooled peptides and 1 μg/ml anti-CD28 antibody for 6 hours. IFN-γ spots were developed using AP Conjugate Substrate Kit (Bio-Rad) and counted on a Immuno-Spot S6 Analyzer (CTL).

## Digital PCR for detection of the type 2 defective mutants

Genomic DNAs were extracted from HTLV-1-infected patients PBMCs. The genomic DNA was adjusted so that the expected number of proviral copies exceeds 1000. Digital PCR was performed using QX 200 Droplet digital PCR system (BIO RAD). The primers and probes were as follows; 5'LTR primers; 5'- CGGAGCCAGCGACAG -3' (sense) and 5'- CCCATTGCCTAGG GAATAAAGG -3' (anti-sense); 5'LTR probe; 5'- CACTCTCCAGGAGAGAA -3'; pX primers; 5'- ACGGCGCTCCTGCTCTT -3' (sense) and 5'- ATTGCTGAGTATTTGAAAAGGAAGG -3' (anti-sense); pX probe; 5'- TCCTGCGCCGTGCC -3'. The 5'LTR probe was labeled with FAM and the pX probe was labeled with VIC. The amplification conditions were 95°C for 10 min and 40 cycles of 30 seconds at 94°C followed by 120 seconds at 53°C, then 98°C for 10 min. The percentage of type provirus was calculated with follows: (1–5'LTR/pX) x 100.

## Supporting information

**S1 Fig. Effects of anti-CD16 antibody to proviral load and clonality in an STLV-1 infected Japanese macaque.** Anti-CD16 antibody was administered into a STLV-1 infected Japanese macaque. (A) To avoid blocking of CD16 detection by the administered antibody, NK cells (CD3⁻CD159A⁺ lymphocytes) were detected before and after administration of anti-CD16 antibody. (B) Proviral load of PBMCs is shown. The time of administration of antibody is indicated by the arrows.
(TIF)

**S1 Table. HTLV-1 integration sites of the seroconverted case and the donor.** This Table presents all integration sites of HTLV-1 provirus of the HTLV-1 seronegative case who received live liver transplantation from the seropositive donor, and the donor. (XLSX)

**S2 Table. HTLV-1 integration sites in cases who received hematopoietic stem cell transplantation.** All integration sites of HTLV-1 provirus at different time point in three ATL patients who received the hematopoietic stem cell transplantation were shown. (XLSX)

**S3 Table. STLV-1 integration sites in a seropositive Japanese macaque treated by anti-CD8 antibody.** This Table presents all integration sites of STLV-1 provirus in an STLV-1 infected Japanese macaque treated by anti-CD8 monoclonal antibody. (XLSX)

**S4 Table. HTLV-1 integration sites in an ATL case treated by Tax-DC vaccine.** All integration sites of HTLV-1 provirus at different time point in an ATL case treated by Tax-DC vaccine were shown. (XLSX)

## Acknowledgments

We thank C. Onishi and R. Furuta for valuable help for experiments.

## Author Contributions

**Conceptualization:** Mikiko Izaki, Jun-ichirou Yasunaga, Masao Matsuoka.

**Data curation:** Mikiko Izaki, Kisato Nosaka.

**Formal analysis:** Mikiko Izaki, Jun-ichirou Yasunaga, Kisato Nosaka, Kenji Sugata, Takafumi Shichijo, Asami Yamada, Anat Melamed, Masao Matsuoka.

**Funding acquisition:** Masao Matsuoka.

**Investigation:** Mikiko Izaki, Jun-ichirou Yasunaga, Hirofumi Akari, Masao Matsuoka.

**Methodology:** Mikiko Izaki, Kenji Sugata, Takafumi Shichijo, Asami Yamada, Anat Melamed.

**Project administration:** Jun-ichirou Yasunaga, Masao Matsuoka.

**Resources:** Kisato Nosaka, Hayato Utsunomiya, Youko Suehiro, Yasuhiko Sugawara, Taizo Hibi, Yukihiro Inomata, Hirofumi Akari.

**Supervision:** Hirofumi Akari, Charles Bangham, Masao Matsuoka.

**Validation:** Mikiko Izaki.

**Writing – original draft:** Mikiko Izaki, Jun-ichirou Yasunaga, Hirofumi Akari, Anat Melamed, Charles Bangham, Masao Matsuoka.

**Writing – review & editing:** Jun-ichirou Yasunaga, Charles Bangham, Masao Matsuoka.

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
