## [Decision Letter · Decision Letter 0]

13 Oct 2020

Dear Prof. Matsuoka,

Thank you very much for submitting your manuscript "In vivo dynamics and adaptation of HTLV-1 infected clones under different clinical conditions" for consideration at PLOS Pathogens. As with all papers reviewed by the journal, your manuscript was reviewed by members of the editorial board and by several independent reviewers. In light of the reviews (below this email), we would like to invite the resubmission of a significantly-revised version that takes into account the reviewers' comments.

We cannot make any decision about publication until we have seen the revised manuscript and your response to the reviewers' comments. Your revised manuscript is also likely to be sent to reviewers for further evaluation.

Sincerely,

Daniel C. Douek

Associate Editor

PLOS Pathogens

Richard Koup

Section Editor

PLOS Pathogens

Kasturi Haldar

Editor-in-Chief

PLOS Pathogens

orcid.org/0000-0001-5065-158X

Michael Malim

Editor-in-Chief

PLOS Pathogens

orcid.org/0000-0002-7699-2064

Reviewer's Responses to Questions

**Part I - Summary**

Reviewer #1: The group is one of the global leaders in the HTLV-1 research field. They addressed a central question when fighting against the fatal leukemia (ATL) caused by this virus; how host innate and adaptive immunity (those by NK and CTLs) control the status of HTLV-1 infection, in particular, the numbers and clonality of HTLV-1+ CD4 T-cells in vivo. This issue has not been addressed before, thus the study presents a significant advancement to the research on HTLV-1 as well as to improve the immune therapy on various HTLV-1 associated diseases. The study encompasses human patients (with some of them treated by the DC vaccination against Tax) and non-human primates that have been naturally infected by the simian leukemia virus-1 and manifesting diseases similar to HTLV-1 infected humans.

There are a few important take home messages that are presented in this article, i.e., the CD8-mediated host immunity has differential effects on the number and clonality of infected cells. In addition, immunity against Tax and HBZ shows different impacts on the homeostasis/expansion of infected cells and on the development of ATL.

Reviewer #2: In this manuscript, Izaki et al report the outcomes of HTLV-1 infection in HTLV-1-negative organ transplant recipients who received either a liver transplantation or allogeneic (?) hematopoietic stem cells transplantation (HSCT) from HTLV-1+ donors. STLV infection in a Japanese Macaques was also analyzed. Overall, the data presented are potentially interesting, but incomplete. And more seriously, the interpretations of data are not consistent with the clinical conditions of the infected transplant recipients who underwent immune-suppressive treatments prior to and throughout the transplantation and infection. It is curious that the authors kept emphasizing the importance of immune surveillance against HTLV-1 when the infections occurred in individuals whose immune cells, especially CD8+ cytotoxic T lymphocytes, were severely depleted by immunosuppressive treatments. For infections that occurred after HSCT, many variables were poorly defined. For examples, it is unclear whether bone marrows or purified CD34+ stem cells were transplanted. The proviral loads in the transplants and the numbers of susceptible CD4+ T cells in the recipients were not known. The lack of critical details makes the collected data difficult to evaluate. Finally, the study did not take into account a body of published literature regarding the outcomes of HTLV-1 infection in cell culture and in animals (e.g., the rabbit model). As a result, some of the conclusions of the study appear inconsistent with the data presented and may be misleading.

**Part II – Major Issues: Key Experiments Required for Acceptance**

Reviewer #1: Not applicable

Reviewer #2: 1. In Fig. 1, the immune-suppressive treatments (Fig. 1A top, mycophenolate, prednisolone, tacrolimus) presumably suppressed the recipient’s CTL response since the transplanted liver seemed tolerated up to 15 months. If so, the HTLV-1-infected T cells from the donor should have been immune tolerated as well. As such, it is erroneous and misleading to conclude that HTLV-1-infected clones were turning over in the transplant recipient because of host immune surveillance (lines 111-116). For this conclusion/ suggestion to be drawn/made, a direct demonstration of increased anti-HTLV-1 CTL response in the transplant recipient is needed. However, in view of the immunosuppressive treatments, such evidence is unlikely to be found. A more likely scenario is outlined below in comment 2 for the authors to consider.

2. The most likely interpretation of the data in Fig. 1 is that the cytopathic effects of HTLV-1, not immune surveillance, caused the death of most newly infected T cells by inducing cell cycle arrest, senescence or apoptosis as has been extensively documented in the literature (which should be cited, by the way). This then led to the rapid turn-over of the infected cell population initially, and selected for latently infected cell clones that expressed viral antigens at low levels or intermittently.

3. For Fig. 2, the rationale and experimental details for the HSCT study are not clear, making the interpretation of the results difficult to understand. What were the conditioning regimens used for the HSCT recipients? Was the HSCT allogeneic? Were bone marrows or CD34+ stem cells used in the transplantation? Clinical protocols/treatments should be described and substantively discussed for proper understanding and interpretation of the data.

4. In Fig. 2, were infected CD4+ T cells or stem cells transplanted into recipients? What were the PVLs in the transplants? What were the numbers of susceptible CD4+ cells throughout the study? Were the hematopoietic systems fully reconstituted after the transplantation? The newly reconstituted hematopoietic systems in the HSCT recipients were expected to reflect those of the donors. As such, it is perhaps not surprising that the HTLV-1-infected clones in the recipients were mostly donor-derived. The authors may want to address whether these clones derived from donor T cells or donor stem cells, and to what extent were donor T cells transplanted during HSCT?

5. For the JM experiment, it would be helpful to know when and how the JM under study was infected and for how long.

6. The outcome of HTLV-1 (or HIV, HSV, etc.) infection is a balance between immune surveillance and the intrinsic pathogenicity of HTLV-1 coupled with its ability to integrate and undergo selection. The cytopathic effect of HTLV-1 was completely ignored in this paper (Lines 137-139).

7. The increased turn-over of infected clones after CD8 depletion in JM suggests that there was continual viral reactivation and de novo infection in STLV-infected JM, and many of the infected cells turned over not only because of CTL killing (which was suppressed by the antibody), but most likely because of the cytopathicity caused by the replication. Here again, the reduction in stax mRNA/STLV proviral DNA after CD8 depletion (Fig. 4A right panel) is consistent with the cytopathic effects of the infection, which likely causes DNA damage, cell cycle arrest, senescence, and apoptosis in cells undergoing viral replication, thus leading to the survival of cells that are not replicating HTLV-1 (lower stax/PVL). As mentioned in comment 2, this has been shown by others and should be discussed in the present context with proper citations.

8. In the discussion (lines 289-301), the infected ATL clones that express Tax intermittently are interesting, but these clones had been selected in vivo during chronic infection The data presented in the study seem more consistent with the notion that the majority of HTLV-1-infected cells turned over rapidly even in the absence of immune control, mostly due to the cytotoxic nature of the infection. Again, this point should be discussed.

**Part III – Minor Issues: Editorial and Data Presentation Modifications**

Reviewer #1: Fig 3_ Does the anti-CD8 antibody used for the flow detection bind to the CD8 T-cells despite the presence of the administered CD8 antibody? If they compete for the epitope or cause steric hindrance to each other, then the flow detection should combine/compare indirect staining of the cells bound by the administered CD8 antibody (by using Fluorophore-labeled anti-simian IgG) and direct staining. Though chances are slim that CD8 T-cell depletion did not occur in vivo, that could change the interpretation, if true.

This goes with the CD16 depletion assay.

Alternatively, if CD8 T-cells did not decrease in number, but only their presence was “stealthed”, that should lead to the increase of CD4CD8 double negative cells. Thus, if the ratio of CD4 vs. nonCD4/CD8 did not show overt decrease in the early days of treatment, it would strongly suggest that the observation supports the specific depletion of CD8 T-cells by the antibody treatment.

Page 27 line 470 _ digital detection of “HTLV-1 type 2 provirus” the word type 2 provirus is misleading as it refers to ” the type 2 defective mutants”, but not a new type of HTLV-1.

Reviewer #2: (No Response)

PLOS authors have the option to publish the peer review history of their article (what does this mean?). If published, this will include your full peer review and any attached files.

Reviewer #1: No

Reviewer #2: No
---

## [Decision Letter · Decision Letter 1]

21 Dec 2020

Dear Prof. Matsuoka,

Thank you very much for submitting your manuscript "In vivo dynamics and adaptation of HTLV-1 infected clones under different clinical conditions" for consideration at PLOS Pathogens. As with all papers reviewed by the journal, your manuscript was reviewed by members of the editorial board and by several independent reviewers. The reviewers appreciated the attention to an important topic. Based on the reviews, we are likely to accept this manuscript for publication, providing that you modify the manuscript according to the review recommendations.

Sincerely,

Daniel C. Douek

Associate Editor

PLOS Pathogens

Richard Koup

Section Editor

PLOS Pathogens

Kasturi Haldar

Editor-in-Chief

PLOS Pathogens

orcid.org/0000-0001-5065-158X

Michael Malim

Editor-in-Chief

PLOS Pathogens

orcid.org/0000-0002-7699-2064

Reviewer Comments (if any, and for reference):

Reviewer's Responses to Questions

**Part I - Summary**

Reviewer #1: Since this is a resubmission and I have already sent in all these previously, I skip this part.

Reviewer #2: The authors have largely addressed the comments raised in previous critiques. The following points should also be discussed and addressed before acceptance.

**Part II – Major Issues: Key Experiments Required for Acceptance**

Reviewer #1: None

Reviewer #2: None

**Part III – Minor Issues: Editorial and Data Presentation Modifications**

Reviewer #1: All issued I raised in the past have been addressed.

Reviewer #2: 1. The statement in page 6, lines 116-119 of the revised manuscript � “Thus, newly-infected HTLV-1 clones dynamically changed, suggesting that appropriate HTLV-1-infected clones for survival in vivo are selected under host immune surveillance. It is noteworthy that many donor-derived HTLV-1 cells persisted in the recipient (Fig 1D).” � remains difficult to reconcile and understand. How can the immune surveillance, on the one hand, controlled the newly infected clones in the transplant recipient, and yet leaving donor-derived HTLV-1-infected cells to persist untouched? This point was raised in reviewer 2, comments 2 and 4 of the previous review.

The manuscript can be improved by considering the following possibilities: (i) The newly infected cells turned over rapidly due to the cytopathic effect of HTLV-1, not primarily due to the host immune response, which would have eliminated both newly infected cells and donor HTLV-1 cells. (ii) Alternatively, only a very small fraction of the previously selected donor HTLV-1 cells were capable of virus replication and transmission (through sporadic reactivation?), and as such most of them evade immune detection.

2. The following literature should also be cited and discussed:

(i) Glowacka et al “Delayed Seroconversion and Rapid Onset of Lymphoproliferative Disease After Transmission of Human T-Cell Lymphotropic Virus Type 1 From a Multiorgan Donor”

(https://academic.oup.com/cid/article/57/10/1417/289550)

(ii) Cook et al “Rapid dissemination of human T-lymphotropic virus type 1 during primary infection in transplant recipients”

Both papers suggest that the host immune response to transplant-associated HTLV-1 infection, while present, is not very robust.

(https://retrovirology.biomedcentral.com/articles/10.1186/s12977-015-0236-7).

(iii) An earlier study had shown that the majority of cells newly infected by HTLV-1 in cell culture became arrested in cell cycle progression or senescent, and only a small fraction of cells that expressed Tax at low levels managed to undergo expansion. (https://pubmed.ncbi.nlm.nih.gov/24699669/)

PLOS authors have the option to publish the peer review history of their article (what does this mean?). If published, this will include your full peer review and any attached files.

Reviewer #1: No

Reviewer #2: No
---

## [Editor Report · Decision Letter 2]

4 Jan 2021

Dear Prof. Matsuoka,

We are pleased to inform you that your manuscript 'In vivo dynamics and adaptation of HTLV-1 infected clones under different clinical conditions' has been provisionally accepted for publication in PLOS Pathogens.

Best regards,

Daniel C. Douek

Associate Editor

PLOS Pathogens

Richard Koup

Section Editor

PLOS Pathogens

Kasturi Haldar

Editor-in-Chief

PLOS Pathogens

orcid.org/0000-0001-5065-158X

Michael Malim

Editor-in-Chief

PLOS Pathogens

orcid.org/0000-0002-7699-2064
---

## [Editor Report · Acceptance letter]

25 Jan 2021

Dear Prof. Matsuoka,

We are delighted to inform you that your manuscript, "In vivo dynamics and adaptation of HTLV-1 infected clones under different clinical conditions," has been formally accepted for publication in PLOS Pathogens.

Best regards,

Kasturi Haldar

Editor-in-Chief

PLOS Pathogens

orcid.org/0000-0001-5065-158X

Michael Malim

Editor-in-Chief

PLOS Pathogens

orcid.org/0000-0002-7699-2064